# Cardiac implantable electronic devices' longevity: A novel modelling tool for estimation and comparison

**Pascal Defaye**[1], **Serge Boveda**[2], **Jean-Renaud Billuart**[3], **Klaus K. Witte**[4]*,
**Maria F. Paton**[4]

**1** University Hospital of Grenoble-Alpes, Grenoble, France, **2** Clinique Pasteur, Toulouse Cedex 3, France, **3** MicroPort CRM, Clamart, France, **4** Leeds Institute of Cardiovascular and Metabolic Medicine, University of Leeds, Leeds, United Kingdom

* k.k.witte@leeds.ac.uk

## Abstract

### Aims

Generator longevity is the key issue for patients, and is also important for payers, yet implanters of Cardiac Implantable Electronic Devices (CIEDs) face a challenge when selecting the appropriate device since battery longevity is only known for previous generation devices and whilst projected longevities are available for current devices, these are not in comparable formats. This study presents a new framework that facilitates an estimation of longevities for all CIEDs of both previous and existing generations that could simplify personalization of the device choice.

### Methods

Longevity can be calculated based upon a simple concept entitled the "power consumption index" (PCI = t x I/C, where t is a constant of 1 hour, I is the current required by the device and C, its battery capacity). We retrieved published data from the user manuals of all commonly used pacemakers including single chamber, dual chamber, cardiac resynchronization and leadless devices. C and the components of current I including background current ($I_{background}$) and the pacing current ($I_{pacing}$) were calculated prior to calculation of the PCI for each device. Subsequently, a set of fictitious patient pool conditions via a Monte-Carlo simulation were used to model CIED survival curves which were then compared with real-life data from the Swedish device registry of previous generation CIEDs. Finally, we modeled survival curves for current generation devices using the PCI model.

**Data availability statement:** Third-party data was publicly sourced for this study from the Swedish ICD & Pacemaker Registry (https://www.pacemakerregistret.se/icdpmr/start.do). The authors confirm others can replicate the study findings in their entirety by directly obtaining data from the Swedish ICD & Pacemaker Registry and following information outlined in the Methods section. The authors had no special access privileges that others would not have when attempting to access the minimal data from the Swedish ICD & Pacemaker Registry. All other relevant data for this study are within the paper and its Supporting information files.

**Funding:** The author(s) received no specific funding for this work.

**Competing interests:** P.D. receives grants and honoraria from Medtronic, Boston Scientific, Abbott, and Microport, S.B. is a consultant for Medtronic, Boston Scientific, Microport, JRB is an employee of Microport CRM who supervised Maxime Corneloup as an intern, KKW has received research funding from the British Heart Foundation, the National Institute for Health Research, the Medical Research Council. He has also received grants and honoraria for teaching and consultancy work from Medtronic, Cardiac Dimensions, Novartis, Abbott, BMS, Pfizer, Bayer and has received an unconditional research grant from Medtronic to support a PhD program at the University of Leeds. These competing interests do not alter our adherence to PLOS ONE policies on sharing data and materials.

**Abbreviations:** CIED, Cardiac Implantable Electronic Devices; C, battery capacity; CRT-P, Cardiac resynchronization therapy with pacemaker; I, current drain; $I_{background}$, background current; $I_{pacing}$, pacing current; $I_{remote/IEGM/Algo}$, current for optional settings; $I_{remote}$, current for remote monitoring; $I_{IEGM}$, current for IEGM storage; PCI, Power Consumption Index; RVPa, right ventricular pacing avoidance; Device manufacturers, Abt (Abbott), Btk (Biotronik), Bsc (Boston Scientific), Mdt (Medtronic), Mcp (Microport)

## Results

Using the PCI approach we were able to calculate longevities for all pacemaker devices under a variety of settings. The modeled $I_{background}$ matched the data reported by manufacturers, and, under a variety of settings, regression analysis showed a low average error rate between industry-reported and modelled longevities (ratio: modelled longevity/industry reported longevity −1) = 0.1±4.0% and 0.1±0.7% for previous and existing SR/DR devices, 1.0±5.0% and 0±3.0% for previous and existing CRT-P, and 0±4.0% for leadless pacemakers, respectively).

More than 50% of the PCI and thereby a significant contributor to longevity was accounted for by $I_{background}$. $I_{pacing}$ was the second largest contributor (20% for standard single and dual chamber devices, 30% for CRT-P and 40% for leadless devices). Certain pacing algorithms and IEGM storage considerably impacted specific devices with longevity losses of up to 1 year. The Monte-Carlo analysis demonstrated consistency between projected longevities by the PCI model and real-life data for historical devices and the calculated longevities that stemmed from this were consistent with the real-world data from Sweden.

## Conclusion

The PCI model combining power consumption and battery capacity allows a comparison of longevity across CIEDs and programming options. Such a tool could help implanters improve personalization of device prescription for their patients and payers to make more informed decisions about tailoring device purchases and programming most appropriate for their population.

## Introduction

Longevity of Cardiac Implantable Electronic Devices (CIEDs) is an important issue for patients, who wish to avoid further surgery, and purchasers, who wish to optimize cost-effectiveness, and is therefore a relevant consideration for clinicians. It is appreciated that there are common discrepancies between declared (future) longevities of generators and their subsequent survival curve once implanted [1,2]. Despite calls for more transparency and industry-wide standardized reporting of longevity [3–5], comparisons of longevity between devices and manufacturers in different settings remains challenging. Given that 30–40% of all CIED procedures are generator replacements, there exists the risk of conflicts of interest for both manufacturers, and, in fee-for-service healthcare environments providers, that limit enthusiasm for transparency [6].

Although implanters and their patients appreciate the concept of battery capacity as a prime criteria for device longevity [7] they also recognize that energy drain plays a role [8,9]. However, the potential lifetime of the device is also determined by how energy is stored, and how efficiently it is delivered [10], along with the patient's characteristics, all of which, add to the frustrating situation of complex and

non-standardized user-manual declared longevity, with different programming as baseline across companies, making personalization of generator prescription impossible even in the presence of similar battery capacity. If one could reliably describe consumption and index this to battery capacity and pacing requirements, there remains the possibility of a reliable comparison of devices.

Based upon previous work defining the power consumption of CIEDs [11], which included a calculation for the inverse of longevity, we have developed the Power Consumption Index (PCI) (as defined by $PCI = t \times I/C$ (where t is a constant equal to 1 hour)) that aims to describe the intrinsic power consumption of the overall system (the pacing system coupled with the battery) during a normalized period (1 hour). The reciprocal of the PCI therefore allows a derivation of longevity (Fig 1 and S1 File).

The Power Consumption Index for two illustrative devices. On the Y axis, the PCI index ($t \times I/C$ with $t = 1\,h$) is split according to each current contribution ($t \times I_{background}/C$, $t \times I_{PACING}/C$, etc..). On the Y' axis, the green lines represent the corresponding longevity in years (a nonlinear "1/x scale"; the inverse of the PCI index $t \times I/C$ multiplied by $10^6$ (I is in mA) and divided by 365x24 gives the corresponding longevity (in years).

Hence, the objectives of this study were 1) to create and test a "universal" model, based on the concept of PCI, that offers the opportunity to model longevity for any CIED; 2) to validate this by comparing the modeled survival curves of previous generation CIEDs with real world data and; 3) apply this to predict product survival curves for new generations of CIEDs.

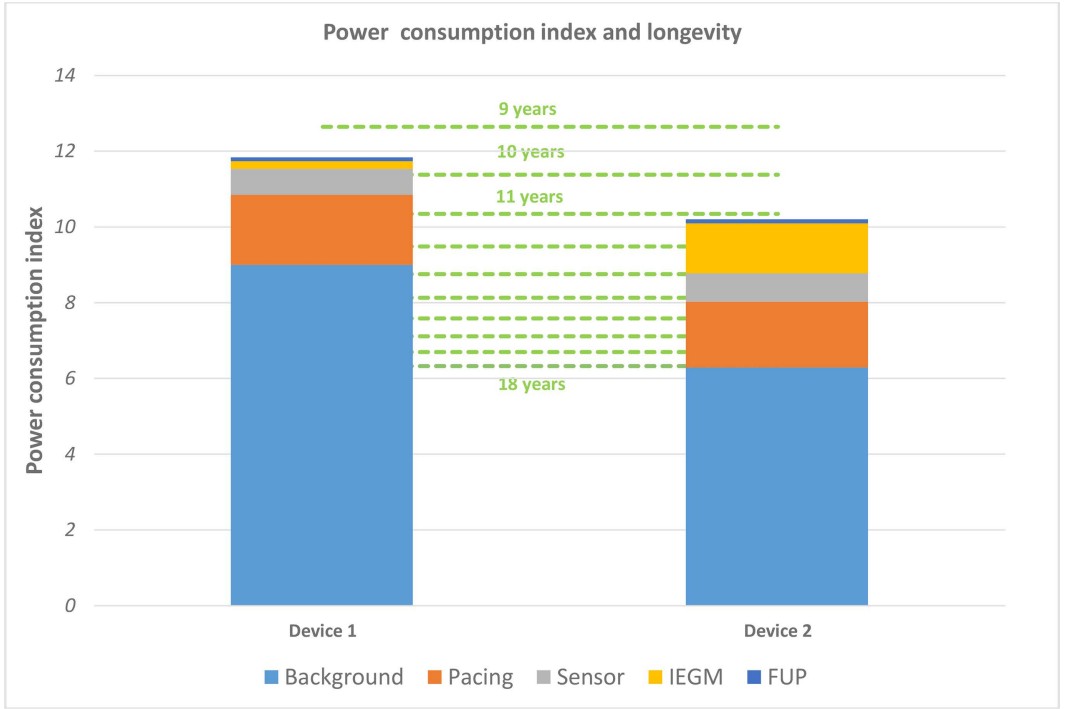

**Fig 1. PCI and nominal longevity.** Note: The Power Consumption Index for two illustrative devices. On the Y axis, the PCI index ($\tau \times I/C$ with $\tau = 1\,h$) is split according to each current contribution ($\tau \times I_{background}/C$, $\tau \times I_{PACING}/C$, etc.). On the Y' axis, the green lines represent the corresponding longevity in years (a nonlinear "1/x scale"; the inverse of the PCI index $\tau \times I/C$ multiplied by $10^6$ (I is in µA) and divided by 365x24 gives the corresponding longevity (in years).

## Methods

Firstly, we collected battery capacity and estimated current drain for a variety of devices from manufacturers' user manuals to calculate PCI values and thereby longevity. The user manuals are available from the five major CIED manufacturers: Abbott (Abt, formerly St Jude Medical, Sylmar, CA, USA), Boston Scientific (BSc, St Paul, MN, USA), Biotronik (Btk, Berlin, Germany), Microport CRM (Mcp, formerly LivaNova and Sorin, Clamart, France), and Medtronic (Mdt, Minneapolis, MN, USA). We also used the only web-based longevity calculator available from BsC [12]. Detailed results are provided in S1 File.

### Battery capacity

Nominal voltage and battery capacities at ERI were systematically retrieved from user manuals for single and dual-chamber (SR and DR) pacemaker, cardiac resynchronization therapy (cardiac resynchronization therapy with pacemaker [CRT-P]) and leadless pacemaker (PM) models. Representative models of previous and new generations of CIEDs were selected based on experts' opinion (Table 1: Device models).

### Current drain modelling

Industry-reported CIEDs' longevities in user manuals depend upon the programming (including the activation of specific algorithms such as rhythm storage, remote monitoring, and sensors). Current is not provided in manuals and therefore modeling was required to calculate the PCI for each CIED. Because nominal settings differ from one manufacturer to another, an additional step was necessary to specifically identify and estimate the current for background device activity and pacing ($I_{background}$, $I_{pacing}$) and optional settings ($I_{remote}$ $I_{IEGM}$ $I_{algo}$). For $I_{background}$, and $I_{pacing}$ the evaluation was done by a regression analysis. For $I_{remote/IEGM/algo}$ the current was estimated via the difference of longevity between activated and deactivated settings for each option.

Two categories of optional settings were considered. The first considered algorithms directly influencing the $I_{pacing}$ drain such as automatic threshold management or reduction of right ventricular pacing (RVP) percentage ($I_{algo}$) and the second explored optional settings such as remote monitoring and IEGM storage ($I_{remote}$ and $I_{IEGM}$). Unlike $I_{background}$ or basic pacing, each factor needed to be analyzed individually (S1 File).

Table 1. Cardiac Implantable Electronic Device models.

| | Previous generations: | Current CIEDs: | Leadless CIEDs |
|---|---|---|---|
| *SR* | Abbott (Identity SR™ Adx Model 5180), Biotronik (Evia SR™), Boston Scientific (INSIGNIA I ULTRA™ - 1190), Medtronic (Adapta SR™, ENPULSE™), Microport CRM (Symphonie SR™), and Vitatron (GR20 SR™). | Abbott (Assurity MRI VR™), Biotronik (Edora 8 VR™), Boston Scientific (Accolade VR™), Abbott (Frontier II 5596, Anthem™ 3112/3212), Medtronic (Azure XT VR™), Microport CRM (Alizea VR™ and Alizea VR remote™) | Abbott (Aveir V™ 1.5V and 2.5V), Medtronic (Micra™ 1.5V and 2.5V) |
| *DDD* | Abbott (Identity XL DR™ Model 5386 and Identity DR™ Model 5380), Biotronik (Evia DR™ and Evia DR-T™), Boston Scientific (INSIGNIA I ULTRA™ - 1291), Medtronic (ENPULSE E2DR™ 31/33, Enrythm DR™, Adapta DR™, ENPULSE E2 DR21™), Microport CRM (Symphonie DR™), and Vitatron (GR70 DR™) | Abbott (Assurity MRI DR™), Biotronik (Edora 8 DR™), Boston Scientific (Accolade DR EL™ and Accolade DR™), Medtronic (Azure XT DR™), Microport CRM (Alizea DR™ and Alizea DR remote™) | Abbott (Aveir V™ 1.5V and 2.5V, Aveir A™ 1.5V and 2.5V) |
| *CRT-P* | Abbott (Frontier II 5596™, Anthem 3112/3212™), Biotronik (Eva HF-T™, Eluna HF™), Boston Scientific (Invive CRT-P W173™), Medtronic (Consulta CRT-P™, Viva CRT-P™, InsyncIII™) | Abbott (Quadra Allure™, Quadra Allure MPP™), Biotronik (Edora HF™, Eluna HF™), Boston Scientific (Visionist™), Medtronic (Percepta CRT-P aCRT™, Percepta CRT-P ™), Microport CRM (Reply CRT-P™) | |

In addition, for the 20 previous generation single chamber and dual chamber pacemakers under investigation, 674 settings were considered, for 11 new generation devices 243 settings were explored and for 8 previous and 5 current CRT-P devices, respectively 294 and 177 settings. Finally for 3 leadless devices 156 settings were considered.

### Estimation of previous devices' nominal longevities, using the PCI model

A literature review was performed to retrieve standard pacemaker programming parameters (for example, heart rate, threshold, pulse amplitude, pulse duration) in clinical studies, registries and clinical practice. Data related to specific algorithms (reduction of ventricular pacing (hysteresis and MVP™, aCRT™), automatic management of pacing output) were also collected via clinical studies, when available. Where the effect on RV pacing of the algorithm was missing, for example with hysteresis and aCRT™, these were imputed using data provided for clinical practice and assumed to be stable over the lifetime of the device. We also assumed that managed output algorithms would achieve and maintain the ideal of 1V. Then, under nominal conditions, the PCIs and the corresponding nominal longevities were estimated for the CIEDs examined (details in S1 File) via the battery capacity values and the current drain modeling using the formulas **PCI=t x I/C** and **L=1/PCI.**

### Validation of PCI model, using Monte-Carlo simulations

In order to validate the survival curves the model was applied not only according to nominal parameters but a variety of settings in order to reflect real-world patient characteristics and programming. A pool of fictitious patient sets (100,000 patients) was created via a Monte-Carlo simulation (programmed in Python™) with age, indication (sinus node dysfunction [SND], intermittent atrio-ventricular block [AVB], complete AVB) and programmed parameters based on available literature [13–20]. The parameters used for the Monte-Carlo model are described in S2 File. Right ventricular pacing avoidance [RVPa] algorithms were assumed to be applied for SND patients eligible for DDD pacing. Remote monitoring was not standard for previous generation devices. We hypothesized a 50% adoption rate of remote for new generation current devices. The impact of settings (such as capture management, additional IEGM storage, RVPa for intermittent AVB for pacemaker, aCRT™ and MPP™ for CRT) on energy consumptions were studied.

For each device, longevity was calculated per patient using via the PCI energy consumption formula. Missing information which could not be derived from manuals was hypothesized, while assuming similarities among same generation devices. When longevity exceeded patient life expectancy, data were censored (since end of service uncommonly occurs simultaneously with death, and residual battery life of the device is rarely collected at death). The distribution of PCI and corresponding longevity across the pool of fictitious patients allowed the drafting of product survival curves for each cardiac implant. The PCI model was then validated for previous generation's devices by comparing these modeled product survival curves and real-life data. For real-world survival curves, we used the Swedish registry [21] which was started in 1989 on the initiative of the Swedish Society of Cardiology. All the implanting clinics in Sweden report to the registry that compiles quarterly and annual reports of pacemaker use in Sweden. Every year there are about 5000 pacemaker procedures in Sweden. The real life product survival curves were extracted from these reports

### Estimation of current devices' longevity, using the PCI model

Finally, the PCI model was used to forecast survival curves for contemporary devices, using the same method as described above.

## Results

### Battery capacity retrieved from manufacturers' manuals

Battery capacities for single and dual chamber pacemakers have remained, on average, unchanged between previous generation and contemporary devices (close to 1 Ah), with disparities among manufacturers (Table 2).

**Table 2. Nominal voltage and capacity by Cardiac Implantable Electronic Device model, according to manufacturers' manuals.**

| | | Previous generation | Nominal voltage (V) | Capacity to ERI (Ah) | | Current generation | Nominal voltage (V) | Capacity to ERI (Ah) | | Leadless pacemakers | Nominal voltage (V) | Capacity ERI (Ah) |
|---|---|---|---|---|---|---|---|---|---|---|---|---|
| **SR** | Abt | Identity™ SR Adx Model 5180 | 2.80 | 0.55 | Abt | Assurity MRI SR™ | 3.20 | 0.91 | Abt | AVEIR LSP201™ Atrial capsule | 3.0 | 0.174 |
| | Btk | Evia™ SR | 2.80 | 1.20 | Btk | Edora 8 SR™ | 3.10 | 0.81 | Abt | AVEIR LSP202™ Ventricular capsule | 3.0 | 0.241 |
| | BSc | INSIGNIA™ I ULTRA-1190 | 2.80 | 0.97 | BSc | Accolade VR™ | 2.80 | 1.00 | Mdt | MICRA™ | 3.2 | 0.12 |
| | Mcp | Symphonie™ SR | 2.80 | 0.93 | Mcp | Alizea SR™ | 3.10 | 1.12 | | | | |
| | Mdt | Enpulse ™ E2SR/Adapta SR | 2.80 | 0.86/0.86 | Mdt | Azure XT SR™ | 3.25 | 0.97 | | | | |
| | Vit | G20 SR | 2.80 | 0.86 | | | | | | | | |
| **DR** | Abt | Identity™ D 5380/DR | 2.80 | 0.55/0.95 | Abt | Assurity MRI DR™ | 3.20 | 0.91 | | | | |
| | Btk | Evia™ DR 2 models DR-T/DR | 3.1/2.8 | 1.05/1.2 | Btk | Edora 8 DR™ | 3.10 | 0.81 | | | | |
| | BSc | Insignia™ ULTRA 1290/1291 | 2.80 | 0.935/1.44 | BSc | Accolade™ DR, DR-EL | 2.80 | 1/1.6 | | | | |
| | Mcp | Symphonie™ DR | 2.80 | 0.93 | Mcp | Alizea DR™ rem/no rem | 3.10 | 1.04/1.12 | | | | |
| | Mdt | Enpulse™21/Adapta™/Enpulse™33 | 2.80 | 0.82/1.2/1.4 | Mdt | Azu XT DR™ | 3.25 | 0.97 | | | | |
| | Mdt | Enrythm™ DR | 3,20 | 1.10 | | | | | | | | |
| | Vit | G70™ DR | 2.80 | 1.22 | | | | | | | | |
| **CRT-P** | Abt | Anthem™ 3112–3212/Frontier™ II | 3.2/2.8 | 0.8/0.95 | Abt | Quadra Allure™ | 3.2 | 0.8 | | | | |
| | Btk | Evia™ HF-T/Eluna™ HF | 3.1 | 1.0 | Btk | Edora HF™ | 3.1 | 0.932 | | | | |
| | BSc | Invive™ CRTP W173 | 3 | 1.36 | BSc | Visionist™ | 2.8 | 1.5 | | | | |
| | Mdt | Insync III™ | 3.25 | 1.4 | Mcp | Reply CRT-P™ | 2.8 | 0.82 | | | | |
| | Mdt | Consulta CRT-P™/Viva CRT-P™ | 3.2 | 0.97 | Mdt | Percepta CRT-P™ | 3.25 | 1.03 | | | | |

Note: Btk is the only supplier pursuing multiple battery sources, but according to the Btk user manuals, differences in battery capacity have no impact on device longevity.

In CRT-P devices, most manufacturers use the same battery capacity as for their conventional pacemaker, close to 1 Ah. Only BSc offers a battery above 1 Ah (using the same technology as Accolade™ DR EL) with 1.5 Ah at elective replacement indicator (ERI).

Battery capacity for single and dual chamber leadless pacemakers is very different from standard pacemakers. For VVI pacing, the Micra™ SR capsule is equipped with a 0.12 Ah battery, while the Aveir™ capsule battery is 0.24 Ah. Aveir™ is also available in a dual chamber configuration with the same ventricular capsule combined with a smaller atrial capsule (equipped with a 0.17 Ah battery capacity).

## Current drain modeling

***Background current (Ibackground) and pacing current (Ipacing).*** For conventional pacemakers, the difference between industry-reported longevities and those derived by regression was 0.1 years±4% for previous generation

devices and −0.1 years ± 0.7% for new generation devices. For devices with a variety of configurations (Mdt and BSc via its longevity calculator website), the regression coefficient ($R^2$) exceeded 90%, across all settings applied.

The $I_{background}$ derived by regression analysis (Table 3) matched those reported by manufacturers for most devices with few exceptions. There was, however, a significant change between previous and new generation devices. Previous generation devices relied on a $I_{background}$ exceeding 9–10 µA (except for Identity™ for which the $I_{background}$ ranged between 5.72 µA and 6.19 µA, the Evia™-T for which the $I_{background}$ ranged between 6 µA and 6.66 µA and Symphony™ with an $I_{background}$ at 6µA). For new generation CIEDs, the $I_{background}$ did not exceed 7 to 7.5 µA, except for BSc devices, which reached 9.7 µA to 10.4 µA. Similar results were observed for CRT-P. For leadless pacemakers, the Ibackground was much lower than for conventional pacemakers (ranging from 0.8 µA to 0.94 µA for SR and 1.75 µA for DR devices, including 0.81 µA to 1µA for communication between the atrial and the ventricular capsules).

The analysis conducted in S1 File *(Pacing current)* shows that for the following settings (60 bpm, pulse width 0.4 ms, lead impedance 500 ohms, with 100% pacing), the $I_{pacing}$ was relatively consistent across all categories of contemporary pacemakers with average current drains of 1.98 ± 0.12µA at 2.5V pacing output and 4.37 ± 0.24 µA, at 3.5V.

***Current from optional settings (Ialgo/remote/sensorIEGM).*** Reduction of ventricular pacing algorithms (RVPa) or Adaptive CRT (aCRT™) directly modulates the percentage of RV pacing. Depending on the amount of pacing avoided, $I_{pacing}$ decreases from 1.98–4.37 µA, respectively at 2,5V and 3,5V (100% pacing), down to 0,89−1,97µA (45% of pacing) and 0,59−1,31µA (30% of pacing). User manuals report that these savings are achieved without energy cost. On the other hand, multipoint point pacing for CRT (MPP™) increases $I_{pacing}$ on the left ventricular channel up to 3,96 and 8,74µA respectively at both 2,5V and 3,5V.

Threshold algorithms whose sole objective is to guarantee capture, increase pacing outputs and pacing current (user manuals do not specify impact on longevity). Other automatic threshold algorithms aim at optimizing outputs and do this either daily (Capture Management™ from Mdt and Capture Control™ from Btk) or on beat-to-beat basis (Auto capture™ from Abt, Automatic capture™ from Bsc). User manuals do not report an energy cost for the daily algorithms but an energy cost of 1 µA can be derived for beat-to-beat algorithms. The analysis conducted for BsC device (see S1 File*: Algorithms influencing pacing current)* shows that the beat-to-beat algorithm saves energy (current drain) only if the percentage of pacing is high (>60%) or outputs exceed 2,5V when the algorithm is deactivated.

Rate adaptive pacing usually relies on a G-sensor (accelerometer) to adapt pacing rate according to effort. Adaption of pacing rate can be optionally enhanced with the combination of a minute ventilation (MV) sensor. User manuals describe an estimation of the energy consumption by the MV sensor of (0.69 µA- 0.77 µA). The impact of rate adaptive technology on pacing is unknown and somewhat unpredictable.

For most suppliers, IEGM storage is embedded as a standard function and the energy cost related to EGM is already included in the current background. Only Mdt reports a specific impact on longevity (*see* S1 File*: IEGM*). While 6-month storage has minimal effects (0,11–0,34 µA for Enpulse™, 0,04–0,11 µA for Azure™), the optional additional use of pre-arrhythmia EGM storage, increases current drain and reduces projected service life by approximately by 34% or 4 months per year for Enpulse (equivalent to 5,67−6,16 µA for Enpulse™ 1,3–1,7µA for Azure™).

For remote monitoring, assuming 2−4 transmissions per year, the current consumption is around 1,14−1,75 µA for RF solutions (BsC, Btk) while it is 0,09−0,59 µA for Bluetooth connectivity (Mdt, McP). Btk provides a unique solution as its devices transmit data daily (alerts are managed via its website) with a fixed energy cost close to 1,75 µA.

## Estimation of nominal longevities, using the PCI model

After deriving current drain, the PCIs were computed and corresponding longevities were modelled, at nominal settings, across all devices. Standard settings, PCI value per device and corresponding longevity, evolution between previous and new generation devices, as well as a sensitivity analysis are reported in S2 File. Fig 2 compares each device per category

**Table 3. Background currents (from manufacturer's manuals and modelling) by device.**

| | | Previous generation CIED models | Reported background current (µA) according to manuals | Modeled background current (µA) | | Current CIED models | Reported background current (µA) according to manuals | Modeled background current (µA) | | | Leadless pacemakers | Background current (µA) from manuals | Modeled background current (µA) |
|---|---|---|---|---|---|---|---|---|---|---|---|---|---|
| **SR** | Abt | Identity™ SR Adx Model 5180 | 6.30 | 5.94 | Abt | Assurity™ MRI SR | 5.40 | 5.70 | **SR** | Abt | AVEIR™ LSP201Atrial capsule | 0.94 | 1.06 |
| | Btk | Evia™ SR | 6.00 | 7.21 | Btk | Edora™ 8 SR | 6.00 | 5.20 | | Abt | AVEIR™ LSP202Ventr. Capsule™ | 0.94 | 1.02 |
| | BSc | INSIGNIA I ULTRA – 1190 | NA | 10.72 | BSc | Accolade™ VR | NA | 9.70 | | Mdt | MICRA™ | 0.8 | 0.94 |
| | Mcp | Symphonie™ SR | NA | 5.78 | Mcp | Alizea™ SR | 5.73 | 5.80 | **DR** | Abt | AVEIR™ LSP201Atrial capsule | 1.8 | 2.03 |
| | Mdt | Enpulse™ E2SR/ Adapta SR | 11/12.93 | 11.1/11.34 | Mdt | Azure™ XT SR | 6.71 | 6.10 | | Abt | AVEIR™ LSP202Ventr. capsule | 1.8 | 1.94 |
| | Vit | G20™ SR | 9.80 | 9.14 | | | | | | | | | |
| **DR** | Abt | Identity™ D 5380/ DR | 6.90 | 6.19/5.72 | Abt | Assurity™ MRI DR | 7.70 | 7.50 | | | | | |
| | Btk | Evia™ DR 2 models DR-T/ DR | 6.00 | 6.66/9.86 | Btk | Edora™ 8 DR | 6.00 | 6.00 | | | | | |
| | BSc | Insignia™ ULTRA 1290/ 1291 | NA | 11.62/13.06 | BSc | Accolade™ DR, DR-EL™ | NA | 10.3/10.4 | | | | | |
| | Mcp | Symphonie™ DR | NA | 6.00 | Mcp | Alizea™ DR rem/ no rem | 6.00 | 6.30 | | | | | |
| | Mdt | Enpulse™ 21/ Adapta™/ Enpulse™ 33 | 13.3/13.87/13.3 | 13.28/14.67/13.7 | Mdt | Azure™ XT DR | 6.71 | 7.00 | | | | | |
| | Mdt | Enrythm™ DR | 9,80 | 9,80 | | | | | | | | | |
| | Vit | G70™ DR | 10,00 | 11,42 | | | | | | | | | |
| **CRT-P** | Abt | Anthem™ 3112– 3212/ Frontier II | NA | 7,09 | Abt | Quadra Allure™ | NA | 7.09 | | | | | |
| | Btk | Evia™ HF-T/ Eluna HF | 7.0 | 7,26/7,56 | Btk | Edora HF™ | 7.0 | 7.1 | | | | | |
| | BSc | Invive™ CRTP W173 | NA | 11,77 | BSc | Visionist™ | NA | 9.63 | | | | | |

*(Continued)*

**Table 3.** (Continued)

| | Previous generation CIED models | Reported background current (µA) according to manuals | Modeled background current (µA) | | Current CIED models | Reported background current (µA) according to manuals | Modeled background current (µA) | | | Leadless pacemakers | Background current (µA) from manuals | Modeled background current (µA) |
|---|---|---|---|---|---|---|---|---|---|---|---|---|
| Mdt | Insync™ III | 12.0 | 11,90 | Mcp | Reply CRT-P™ | 7.9 | 7.5 | | | | | |
| Mdt | Consulta™ CRT-P/ Viva CRT-P | 7.07/7.22 | 7,06/7,22 | Mdt | Percepta CRT-P™ | 7.14 | 7.19 | | | | | |

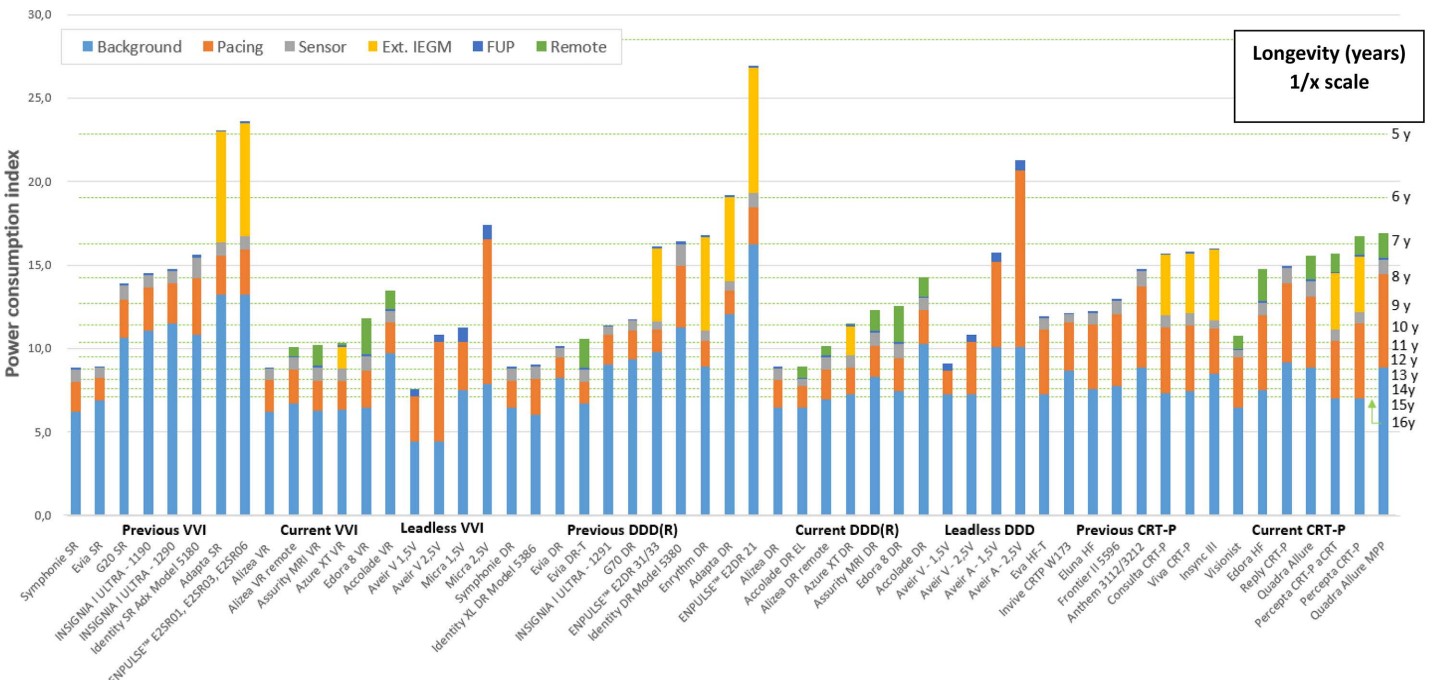

**Fig 2. PCI and longevity with nominal settings for pacemaker devices.** The Power Consumption Index and longevity of previous and current devices describing the contribution of each setting. For conventional pacemakers, the settings considered were: basic rate: 60bpm, pacing threshold: 2.5V, pulse duration: 0.4ms and impedance: 500 ohms for both A&V. For VVI pacemakers, ventricular pacing: 90%. For dual chamber pacemakers, atrial pacing: 70% for SND, 30% for AVB (51% on average) and ventricular pacing assumptions accounted for the difference between AAI/DDD mode and other RVP algorithms (29% vs 47%). Options such as sensor, IEGM storage and remote monitoring (2 transmissions/year) are reported. For leadless VVI and DDD pacemakers, the settings used were: basic rate: 60bpm, pulse duration: 0.25ms for Micra™ and 0.4ms for Aveir™, impedance: ~600 ohms for ventricle and ~300 ohms for the atrium. Pacing outputs were not reported in studies and two options were considered: 1.5 V or 2.5V reflecting the level of confidence of practitioners in adapting output (thresholds observed were typically low: 1.25 V at implant and 0.75 V weeks after). Pacing percentages were the same as for conventional pacemakers. Hysteresis mode was applied for DDD. For CRT-P, BIV was the standard pacing mode (60bpm, 50% A, 100% BIV, 500 ohms) with alternative options such as aCRT™ or MPP™ pacing. Ext. IEGM: extended IEGM, FUP: interrogation of cardiac implant via inductive during a face to face follow-up (one per year).

(SR, DR, CRT-P) according to the PCI chart presented previously. Across all devices, the PCI values range from 26.9 (corresponding to a longevity of 4.2 years) (Enpulse) and 7.6 (with an estimated longevity of 15.1 years) for the single chamber Aveir ventricular device if the programmed output is set to 1.5V).

The Power Consumption Index and longevity of previous and current devices describing the contribution of each setting.

For conventional pacemakers, the settings considered were: basic rate: 60bpm, pacing threshold: 2.5V, pulse duration: 0.4ms and impedance: 500 ohms for both A&V. For VVI pacemakers, ventricular pacing: 90%. For dual chamber pacemakers, atrial pacing: 70% for SND, 30% for AVB (51% on average) and ventricular pacing assumptions accounted for the difference between AAI/DDD mode and other RVP algorithms (29% vs 47%). Options such as sensor, IEGM storage and remote monitoring (2 transmissions/year) are reported. For leadless VVI and DDD pacemakers, the settings used were: basic rate: 60bpm, pulse duration: 0.25ms for Micra™ and 0.4 ms for Aveir™, impedance: ~600 ohms for ventricle and ~300 ohms for the atrium. Pacing outputs were not reported in studies and two options were considered: 1.5 V or 2.5V reflecting the level of confidence of practitioners in adapting output (thresholds observed were typically low: 1.25 V at implant and 0.75 V weeks after). Pacing percentages were the same as for conventional pacemakers. Hysteresis mode was applied for DDD. For CRT-P, BIV was the standard pacing mode (60bpm, 50% A, 100% BIV, 500 ohms) with alternative options such as aCRT™ or MPP™ pacing. Ext. IEGM: extended IEGM, FUP: interrogation of cardiac implant via inductive during a face to face follow-up (one per year).

On average, the PCI for conventional pacemakers is lower for current generations compared with previous generations leading to an increase in longevity for both SR and DR devices (10.8 years *vs.* 15.4 years for SR and 11.2 years *vs.* 14.3 years for DR). Unlike standard pacemakers, the average PCI for CRT-P increased (from 12.5 to 14.1) with the introduction of remote monitoring leading to a reduction of longevity (from 8.3 years to 7.8 years). For leadless devices, the PCI reached 14.2 (corresponding to a longevity of 8.8 years) for dual chamber and 11.7 (10.6 years) for single chamber devices demonstrating the consequence of energy cost of transmission between capsules in the two-chamber system.

The split of PCI per current highlights a strong impact of the $I_{background}$ for all categories of pacemaker: more than 50% of PCI is due to $I_{background}$. The reduction of total PCI for SR/DR between previous and new generations of conventional CIEDs resulted primarily from the reduction of the $I_{background}$. For conventional pacemakers, $I_{pacing}$ accounted for only 20% of the total PCI for SR/DR pacemakers and for 30% of the PCI for CRT-P.

Among the contemporary devices, Accolade™ SR/DR had the highest PCI, and thus the lowest estimated longevity (PCI of 13.5 and longevity of 8.5 years for SR, PCI of 14.2 and longevity of 8 years for DR). This is because this device had the highest $I_{background}$ (10μA) as compared with other devices in the same category. On the other hand, Accolade™ DR EL, even with activated remote monitoring, had the lowest PCI (8.9), and thus the longest longevity (12.8 years), thanks to the high battery capacity at 1.6Ah.

For the other devices, differences were primarily due to optional settings such as extended IEGM, sensor or remote monitoring. In the past, IEGM storage negatively impacted Mdt device longevity. This has been significantly improved upon for standard pacemakers. Moreover, remote monitoring power consumption is three times higher for RF solutions than Bluetooth solutions (PCI for remote monitoring 1.2 vs 0.4). For example, the estimated longevity for Edora™ 8 reached 9.7 years for SR and 9.1 years for DR. Azure™ benefits from a low remote monitoring power consumption from Buetooth and reached 12.7 years for SR and 11.7 years for DR (extended IEGM turned "Off"). Alizea™ SR and DR benefit from additional battery capacity especially if remote monitoring is switched off, such that nominal longevity reached 12.9 years, similar to that of the Accolade™ DR EL device. The impact of activating the G-sensor was not different between devices.

For leadless SR and DR devices, assumptions included a basic rate of 60bpm, a pulse duration of 0,25 ms for Micra™ (Mdt) and 0,4 ms for Aveir™ (Abt), an impedance of ~600 ohms for ventricular pacing and ~300 ohms for atrial pacing [22–25]. Pacing outputs were not reported in studies and two options were considered: 1.5V or 2.5V, reflecting the level of confidence of practitioners in adapting output (thresholds observed were typically low: 1.25 V at implant and 0.75 V weeks after). Pacing percentages were the same as the one for conventional pacemakers. Hysteresis mode was applied for DDD.

Unlike conventional pacemakers, PCI related to pacing in leadless pacemakers accounted for more of the total PCI (40% on average, higher with 2.5V outputs). This is the consequence of a lower battery capacity and thus, a greater proportion of total available energy is required for pacing. Consequently, PCI and longevity significantly changed depending on pacing output assumptions (Aveir™ SR: 15.1 years if at 1.5V *vs.* 10.6 years if at 2.5V; Micra™ SR:10.2 years *vs.* 6.6 years, respectively; Aveir™ ventricular device (DDD mode): 12.5 years *vs.* 10.6 years; Aveir™ atrial capsule (DDD mode) 7.3 *vs.* 5.4 years, respectively).

For CRT-P, biventricular pacing is the standard pacing mode albeit there are additional options such as aCRT™ or MPP™ pacing. Biventricular pacing with the Visionist CRT-P device is associated with a PCI of 10.7 and a longevity of 10.6 years, whereas the Percepta™ device with aCRT activated achieves a PCI of 12.3 and longevity of 9.3 years. Not surprisingly Quadra Allure™ with MPP™ activated suffered a considerable increase in PCI and a corresponding reduction in longevity (PCI: 16.9; longevity of 6.8 years).

Sensitivity analysis of longevity related to fluctuations of currents (see S2 File) reveal a standard deviation close to 3–4% (ratio: sigma divided by nominal longevity) across all devices. For a small, clinically achievable sample size this led to a 95% CI for nominal longevity of 0,3–0,7 years, whereas a simulation of 100 revealed a 95%CI of 0,04–0,10 years and for a simulation of 40,000 the 95%CI was 0,003–0007 years.

## Validation of PCI model, using Monte-Carlo simulations

**Previous generation devices.** Modeled survival curves with standard assumptions fitted Swedish registry data for multiple models with few exceptions (S2 File). For conventional pacemakers, modeled survival curves departed from real-life data for following models: Enrythm™ DR, Evia™ DR-T and Identity™ DR 5370. For Enrythm™, the programming of IEGM storage was the only parameter explaining the difference. For Evia™ DR-T and for Identity™ DR Adx, the difference between modeled survival curves could be explained by the automatic threshold management mode. Overall, the model showed a good fit for CRT-P available in the Swedish registry (InSync III™, Invive™, Frontier II™, Anthem™). This comparison could not be pursued further since real-world programming of implants is not available on the Swedish registry website. Nevertheless, the model showed consistency between real-world and modelled survival curves for most CIEDs from previous generations (Fig 2).

## Estimation of current devices' longevity, using the PCI model

**Conventional pacemakers.** Survival curves for current devices are shown in Fig 3 and reported in S3 File. The 95% confidence intervals are extremely low, due to the size (100,000) of the simulated population, allowing a comparison across devices (S3 File). The aggregated survival curves for conventional pacemakers showed wide differences between devices and manufacturers. Accolade™ SR and DR have the shortest estimated lifespan while the extended longevity DR version of Accolade™ offered the best longevity. The impact of programming options is reported in S2 File.

The figures show the modelled curves for current generation devices.

Among possible settings, the beat-to-beat automatic threshold management algorithm (Auto capture™, Automatic capture™) tended to straighten product survival curve with energy savings on side of the inflexion point and energy cost on the other side, suggesting that optimal programming could extend median product longevity. Reduction of ventricular pacing for intermittent AVB via AAI/DDD mode (available for Btk, Mcp and Mdt devices) extended median longevity for the corresponding devices and reduced the difference between Accolade™ DR EL and Azure™ XT DR or Alizea™ DR. The number of remote transmissions had a marginal effect on longevity. Two devices (Edora™, Alizea™) showed a reduced longevity simply by activation of remote monitoring (by a one-off increase of current for Edora™ or by a reduction of battery capacity for Alizea™).

**Leadless pacemakers.** In the single chamber segment, Aveir™ V significantly outperformed Micra™ pacemaker thanks to its larger battery capacity. Nevertheless, the Micra™ equipped with Capture™ management was associated with a survival curve at 1,5V that matched the Aveir™ survival curve at 2,5V (Autocapture™ is not available on Aveir™).

For two chamber leadless devices, two survival curves were considered, one for each capsule. The Aveir™ atrial capsule had a shorter lifespan compared with the ventricular capsule. The simulation emphasizes the need to optimize pacing outputs for the atrial capsule and the importance of using the RV pacing avoidance algorithm.

**CRT-P.** Of the CRT devices, the modelled survival curves showed a significant difference between Visionist™ and similar devices. Only Percepta™ with the aCRT™ algorithm reached a similar longevity performance. Activation of MPP™ pacing negatively impacts longevity (Fig 3a and b).

## Discussion

The current study firstly describes a novel way to estimate generator longevity by combining current and battery capacity, and then validates this model across a variety of programming options and previous generation of devices by comparing the modeled data with observed longevity from a country-wide registry, and finally provides estimations of the longevities of currently implanted devices for which there are no reliable observed data.

Longevities from user manuals are difficult to use for implant decision making because manufacturers provide these with a variety of settings, pacing options and configurations. In addition, the lack of a common framework does not facilitate an understanding of the determinants of longevity and a comparison between devices. Calculations of longevity not only should use settings reflecting clinical practice but also split power consumption according to unavoidable current usage (background current) and optional algorithms to help practitioners in their implant decision and subsequent programming. We summarize here the key findings and a few recommendations.

### Battery capacity and background current

Battery capacity as a standalone criterion is irrelevant. This study illustrates this using the examples of leadless pacemakers which, despite much lower battery capacity, achieve a reasonable longevity compared with a standard pacemaker. On the other hand, background current plays a key role (at least 50% of PCI is due to $I_{background}$ across all devices). Leadless pacemakers for example, benefit from a substantially lower background current compared with standard pacemakers. Therefore, both battery capacity and background current need to be combined in order to provide the foundation for longevity assessment. This is the purpose of the index proposed here. The PCI combines background current (i.e., $C/(365*24*10^{\wedge -6} I_b)$) and battery capacity and thereby the energy capacity (in years) available for pacing and programming options. Applying this criterion could contribute to personalized device selection, with a focus on device longevity for a wide range of patients (see Table 4).

### Pacing current

The present analysis did not reveal major differences between manufacturers within each category of device, but, by exploring the differences between conventional pacemakers and leadless pacemakers, demonstrated the relative importance of energy consumption and its impact on device longevity as long as lower pacing outputs (<1.5V) can be achieved without loss of capture. As pacing output reaches 2.5V, power consumption increases and longevity is significantly impacted particularly in those devices with lower battery capacity, emphasizing the need to target pacing output close to 1−1.5V. Particularly in leadless dual chamber devices, the atrial capsule which has a modest energy capacity has the potential to limit longevity if atrial pacing is above 80% due to the additional costs of inter-capsule communication. Presently therefore, a dual chamber approach to SND requires careful consideration, especially given the younger age of the affected population. On the other hand, AV Block, with reliable intrinsic atrial activity is likely to be a more useful application for these until further technological developments improve the energetic demand of this connection.

**Current drain from optional programming.** Conventional pacemakers are often chosen specifically for their programming options but the impact of these on longevity is highly variable and depends upon the manufacturer and the device generation. Amongst RVP avoidance algorithms, AAI/DDD mode may be more effective in reducing RV pacing than other algorithms provided it can deal with intermittent AVB.

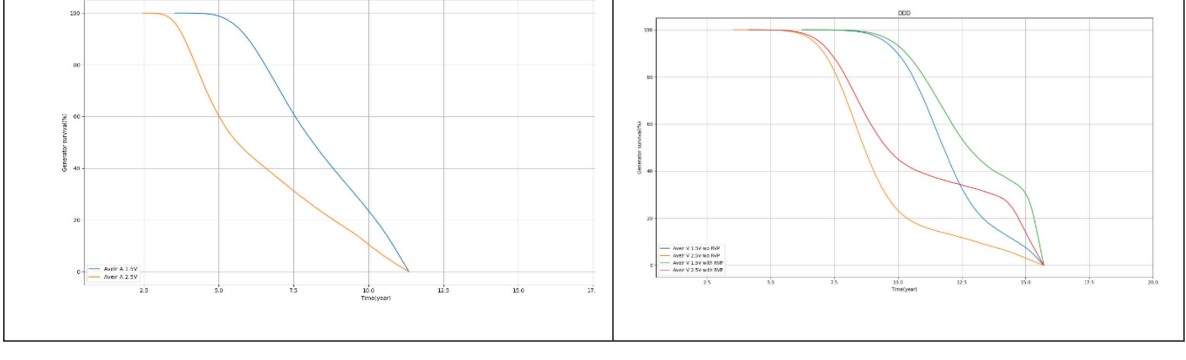

Automatic threshold management has the potential for an adverse effect on battery longevity if the threshold is low and should be deactivated in this situation. The benefit of automatic threshold management is also limited in patients where RVP avoidance is effective [17]. 'Daily' algorithms are also preferable since beat-to-beat algorithms have a larger energy cost. For leadless pacemakers, as longevity is very sensitive to pacing outputs, daily automatic threshold management may be more critical for longevity than for standard pacemakers.

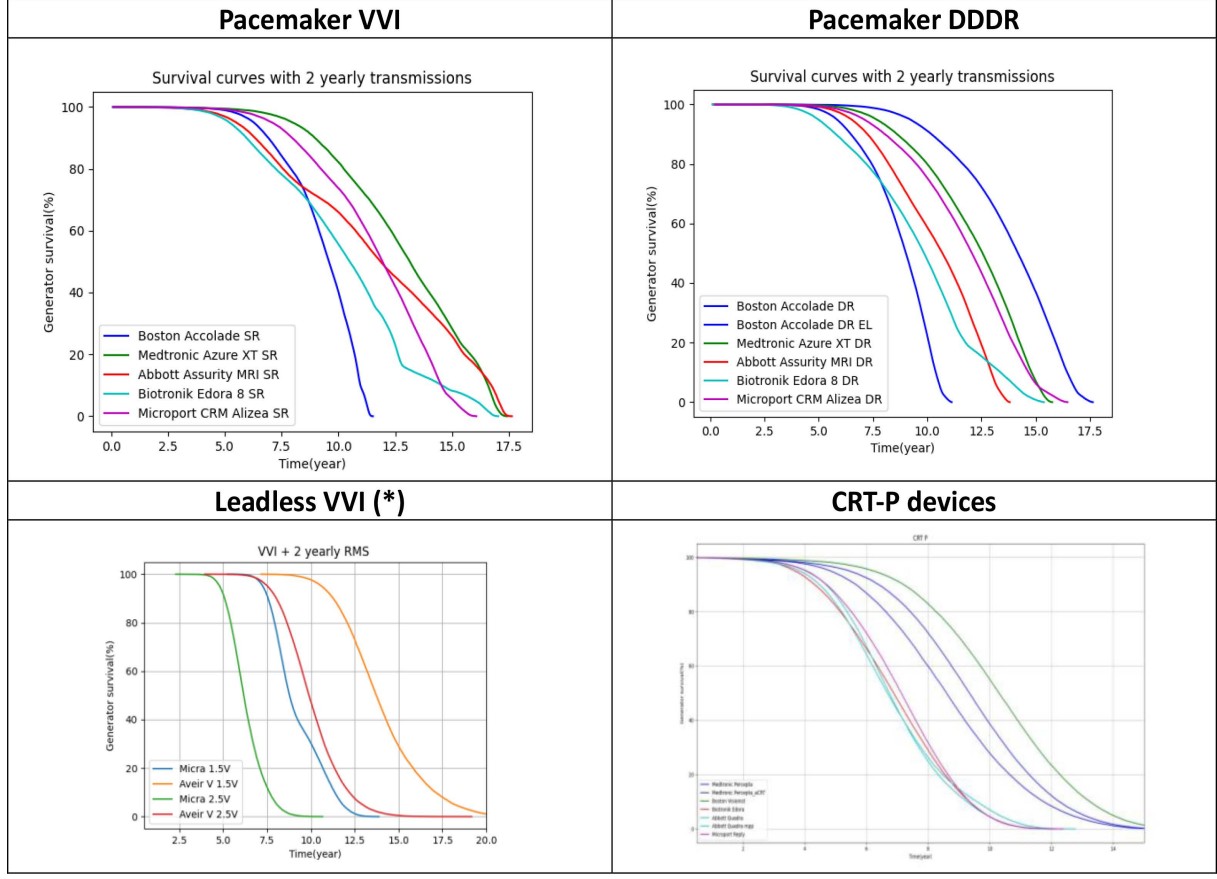

(*) No daily check, 2 Radiofrequency transmissions/year

**Fig 3. Modelled survival curves for current generation devices without (3a) and with (50% remote transmission adoption and 2 yearly transmission) (3b) activated.** (*) No daily check, 2 Radiofrequency transmissions/year. The figures show the modelled curves for current generation devices.

Whether extended IEGM storage and remote monitoring are required should be carefully reviewed on a patient-by-patient basis. For some devices, activation of remote monitoring generates a "one off" energy cost; fortunately, the impact of the number of transmissions on longevity remains marginal.

## Overall PCI and longevity

Overall there has been an increase in longevity of more than one year for conventional SR/DR pacemakers between previous and current generation devices, although there has been a reduction of longevity for CRT-P devices. For leadless devices longevity is similar to that of a standard single chamber pacemaker, but as described, the longevity of two chamber leadless devices is currently predicted to be substantially lower than a standard dual chamber pacemaker. Disparities between suppliers still prevail and should be taken into consideration by clinicians when deciding on device prescription.

## Product survival curves

Our concept was successfully validated using real-life data from previous generation devices, supporting our proposal that the calculated forecasted product survival curves for current devices will be robust.

**Table 4. Recommendations on device selection and programming.**

| Steps | Overall recommendations | Specific points |
|---|---|---|
| 1 | • For device selection, investigate PCI index related to current background (this insures that the device offers the best energy resource for therapy delivery once current background is factored) | • For conventional SR, DR, avoid Accolade™ SR/DR (non EL-version) and for CRT-P, prefer Visionist™, Percepta™ (+aCRT™)<br>• For leadless auricular, anticipate early replacement for SND patients and then prefer leadless DDD for intermittent AVB |
| 2 | • Then, depending on patient profile, select device with adequate pacing options:<br>  • When DDDR is recommended, give priority to reduction of ventricular pacing<br>  • When VVI or CRT-P is required or when percentage of pacing with DR exceeds 60%, check capture algorithm is available to reduce pacing output (in particular for leadless) | • Use available RVPa for SND pts; prefer AAI/DDD for both SNDs & inter. AVB<br>• RV capture is inefficient if combined with aCRT™ |
| 3 | • Check need of additional options (IEGM additional storage, sensors..) | • Among options additional IEGM storage impacts the most longevity |
| 4 | • Carefully program remote follow-up and alert to insure relevant transmissions | • Anticipate "one off" energy cost for some devices (Edora™, Alizea™)<br>• Avoid using inductive telemetry for leadless pacemaker |

### Next steps

In order to provide greater applicability of the PCI concept one could apply it to a prospective cohort where device programming and longevity outcomes are tracked, validating its predictive capacity beyond historical data and, in parallel conduct benchtop measurements for key programming features (e.g., automatic capture management, aCRT) to empirically validate modeled current assumptions. One could analyze how real-time remote monitoring logs correlate with PCI-predicted consumption, offering a feedback loop for future model refinement.

### Limitations

**Current drain modeling.** For some devices, pacing current data could not be obtained for all output values and were hypothesized by assuming similarities among same generation devices. These assumptions were mitigated via an analysis of the relationship between current and outputs highlighting consistency across devices. Options which did not specifically lead to an impact on longevity were assumed to require minimal current (e.g., RVP avoidance algorithms and standard IEGM storage) and are not considered as generating additional energy cost.

**Harmonization of nominal conditions and calculation of nominal PCI: Validation of the model.** The literature review performed to retrieve standard pacemaker parameters provided information on the average settings of devices. As RVP avoidance algorithms were not consistently investigated through clinical trials, percentages of pacing achieved were assumed to be identical according to type (AAI/DDD mode, hysteresis mode such as SAV+°, VIP°) for SND patients. On the other hand, the impact of AAI/DDD modes (MVP°, SafeR°, Vp suppression°) for intermittent AVB patients was investigated in clinical trials and therefore could be considered in the model. For these patients we considered a 1.5% increase per year of ventricular pacing to take into consideration progressive A-V node disease over time. Apart from this group, we had to assume that settings are stable beyond 2–3 years as we lacked clinical data over a longer time frame.

Alternatively, another approach would have been to build a Markov model to incorporate these changes across time, but the probability of transition would have been 'heuristic' since such data are not available in literature. This would have added significant complexity to our model without changing our overall point and making the present manuscript less accessible.

**Modeling survival curves.** The main limitation is the fact that real-life programming of implants was not accessible through the Swedish registry website and hence the model could not be tested with all the combination of options available. The consistency observed between real-life and modelled survival curves suggest that real-life programming does not significantly depart from the settings populated in the model, but this could not be verified.

## Conclusions

Projected longevities of CIEDs are needed for device selection and optimal programming at the time of the implant. Information from user manuals remains difficult to apply, due to a lack of harmonization in the estimated longevities provided by manufacturers regarding settings and programming conditions. We present for the first time, a model based on the Power Consumption Index (PCI) which offers the potential to compare longevity between devices under multiple settings and programming conditions. Longevities estimated by the PCI model appear to be consistent with real-life data for multiple CIED models from different manufacturers. This information could provide implanters, and their patients, the opportunity for personalized pacemaker hardware prescription whilst also paving the way towards standardized reporting of CIED longevity.

### What's new?

The Power Consumption Index (PCI) is a new approach to estimating the longevities of CIEDs across models, manufacturers, settings and pacing options that allows comparisons across devices and manufacturers.

## Supporting information

**S1 File. Model development.**
(DOCX)

**S2 File. Power consumption index rational; Power consumption and nominal longevities; PCI and longevity model sensitivity analysis; Survival curves generated by the Monte-Carlo modelling; Settings and distribution used for Monte-Carlo simulations; Survival curves generated for previous generation devices, Impact of settings for previous generation standard pacemaker devices.**
(DOCX)

**S3 File. Survival curve and confident intervals.**
(XLSX)

## Acknowledgments

The authors thank Ernest W. Lau, MD for insightful discussion related to cardiac device implant longevity and Maxime Corneloup for programming Monte-Carlo simulations with Python™ software. The authors also want to thank the reviewers for their advice during the review process.

## Author contributions

**Conceptualization:** Pascal Defaye, Serge Boveda, Jean-Renaud Billuart, Klaus K. Witte.

**Data curation:** Jean-Renaud Billuart.

**Investigation:** Serge Boveda, Jean-Renaud Billuart, Klaus K. Witte, Maria F. Paton.

**Methodology:** Pascal Defaye, Serge Boveda, Jean-Renaud Billuart, Maria F. Paton.

**Writing – original draft:** Pascal Defaye, Jean-Renaud Billuart, Klaus K. Witte.

**Writing – review & editing:** Pascal Defaye, Serge Boveda, Jean-Renaud Billuart, Klaus K. Witte, Maria F. Paton.

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
