## [Decision Letter · Decision Letter 0]

28 May 2025

PONE-D-25-20064Cardiac Implantable Electronic Devices longevity: A novel modelling tool for estimation and comparison.PLOS ONE

Dear Dr. Witte,

Thank you for submitting your manuscript to PLOS ONE. After careful consideration, we feel that it has merit but does not fully meet PLOS ONE’s publication criteria as it currently stands. Therefore, we invite you to submit a revised version of the manuscript that addresses the points raised during the review process.

We look forward to receiving your revised manuscript.

Kind regards,

Hamed Tavolinejad

Academic Editor

PLOS ONE

Journal Requirements:

I have read the journal's policy and the authors of this manuscript have the following competing interests:  P.D. receives grants and honoraria from Medtronic, Boston Scientific, Abbott, and Microport

S.B. is a consultant for Medtronic, Boston Scientific, Microport.

JRB is an employee of Microport CRM who supervised Maxime Corneloup as an intern.

KKW has received research funding from the British Heart Foundation, the National Institute for Health Research, the Medical Research Council. He has also received grants and honoraria for teaching and consultancy work from Medtronic, Cardiac Dimensions, Novartis, Abbott, BMS, Pfizer, Bayer and has received an unconditional research grant from Medtronic to support a PhD program at the University of Leeds.

4. In the online submission form, you indicated that the data underlying the results presented in the study are available from the authors.

5. We note that you have indicated that there are restrictions to data sharing for this study. For studies involving human research participant data or other sensitive data, we encourage authors to share de-identified or anonymized data. However, when data cannot be publicly shared for ethical reasons, we allow authors to make their data sets available upon request. For information on unacceptable data access restrictions, please see http://journals.plos.org/plosone/s/data-availability#loc-unacceptable-data-access-restrictions.

6. Please amend either the title on the online submission form (via Edit Submission) or the title in the manuscript so that they are identical.

7. Please include a separate caption for each figure in your manuscript.

8. Please remove all personal information, ensure that the data shared are in accordance with participant consent, and re-upload a fully anonymized data set.

9. Please include your tables as part of your main manuscript and remove the individual files. Please note that supplementary tables (should remain/ be uploaded) as separate "supporting information" files.

10.Please include captions for your Supporting Information files at the end of your manuscript, and update any in-text citations to match accordingly. Please see our Supporting Information guidelines for more information: http://journals.plos.org/plosone/s/supporting-information. 

Additional Editor Comments :

Thank you for submitting your manuscript to PLOS One. We have now received comments from three reviewers. Based on their feedback and our editorial assessment, we believe your work addresses an important question, is of good quality, and demonstrates methodological rigor. However, several substantive issues have been raised that require your attention. We invite you to submit a revised version of your manuscript that thoroughly addresses all reviewer comments, either through changes to the manuscript or a clear point-by-point response.

We look forward to receiving your revised submission.

Reviewers' comments:

Reviewer's Responses to Questions

**Comments to the Author**

1. Is the manuscript technically sound, and do the data support the conclusions?

Reviewer #1: Yes

Reviewer #2: Yes

Reviewer #3: Yes

2. Has the statistical analysis been performed appropriately and rigorously? 

Reviewer #1: N/A

Reviewer #2: I Don't Know

Reviewer #3: Yes

3. Have the authors made all data underlying the findings in their manuscript fully available?

Reviewer #1: Yes

Reviewer #2: Yes

Reviewer #3: Yes

4. Is the manuscript presented in an intelligible fashion and written in standard English?

Reviewer #1: Yes

Reviewer #2: Yes

Reviewer #3: Yes

5. Review Comments to the Author

Reviewer #1: Thank you for the opportunity to review this manuscript.

The authors present a novel and unified framework for evaluating battery consumption across all CIEDs, introducing the concept of the Power Consumption Index (PCI). This concept is impressive and has the potential to provide valuable insights into the characteristics of individual CIEDs. I would like to raise a few questions for the authors:

#1. In the Introduction, the authors refer to their previous work, which involves the inverse of device longevity, and define PCI as t × l / C. However, this explanation feels somewhat abrupt. Without a careful reading of the supplemental materials, it is difficult to understand the logical basis for this definition.

#2. Figure 2 is particularly impressive, as it demonstrates the variability in PCI for background current and other functions across different pacemaker models. Could the authors comment on the factors that contribute to differences in PCI for background current (I_background) between models, even among those in the same category and generation?

#3. What are the potential clinical implications of the PCI concept for practitioners, beyond the conventional longevity estimates provided by manufacturers during device checks? How should clinicians apply the PCI framework when selecting pacemaker models within the same category?

Reviewer #2: The study by Defaye et al. offers a novel and practical approach to estimating and comparing the longevity of CIEDs using a Power Consumption Index (PCI) derived from battery capacity and modeled current drain. The inclusion of diverse device types and manufacturers, combined with a large-scale Monte Carlo simulation, enhances the generalizability of the findings. Additionally, the validation of model outputs using data from the Swedish device registry further supports the study’s clinical relevance. However, there are several limitations related to the PCI model and the interpretation of its outputs that should be discussed further to improve the robustness and applicability of the findings.

1. A major limitation of the model is that the authors assumed that device settings and current drain remain stable beyond 2–3 years. However, in practice, pacing thresholds, lead performance, and device efficiency can change over time, and battery internal resistance increases as the device ages. These changes are especially relevant in long-lived or high-burden pacing scenarios and can significantly alter energy consumption over the device’s lifespan. The authors need to address how such non-linear and time-dependent variations might affect the validity of their model. Given these limitations, the PCI model may be more appropriate as a comparative framework for evaluating relative differences between devices, rather than as a precise predictor of absolute longevity.

2. The authors estimated background and pacing currents by using regressing based on manufacturer-reported device longevities. However, these reported longevities are often rounded, modeled, or based on internal assumptions that are not publicly transparent. As a result, even small uncertainties in input longevity, such as a difference of ±0.5 years, can lead to noticeable changes in the inferred current drain and the resulting PCI values. I suggest the authors perform a sensitivity analysis to examine how small variations in input longevity values, for example within a range of ±0.5 to ±1 year can impact the inferred current drain and the resulting PCI. This further sensitivity analysis would help quantify the robustness of the model, clarify which device categories or settings are more sensitive to input variability, and improve the overall transparency and reliability of the model.

3. In the Monte Carlo simulation, the authors modeled 100,000 virtual patients with varying physiological and device-related parameters, including pacing burden, output voltage, lead impedance, algorithm usage, and settings such as IEGM and remote monitoring. However, the results are presented only as point estimates, without reporting summary statistics such as standard deviations, interquartile ranges, or 95% CI. I recommend the authors include these metrics to reflect the level of uncertainty in the estimates. This would clarify whether the differences in predicted longevity across devices are statistically meaningful or potentially fall within the range of expected variability. Without this information, comparative interpretations may be misleading.

Reviewer #3: Major Points

1. The concept of PCI presents a clear advancement in the comparative assessment of CIED longevity. By attempting to unify device longevity estimation into a harmonized, simulation-validated model, this work contributes significantly to a longstanding gap in device transparency and comparability.

Suggestion for enhancement: The manuscript would benefit from a deeper contextualization of PCI relative to past methods (e.g., specific critiques of vendor calculators and earlier power consumption models). This would clarify the incremental benefit of PCI beyond “simplification.”

2. The use of Monte-Carlo simulation based on a robust sample (100,000 virtual patients) and comparison with real-world registry data lends credibility to the PCI framework. However, the validation is partially limited due to assumptions required for missing data and registry limitations.

Suggestion: Include a sensitivity analysis of model assumptions (e.g., effects of varying remote monitoring or pacing algorithm adoption rates). This would strengthen confidence in generalizability.

3. The coverage across device types and manufacturers is commendable. The device-level analysis is precise and allows clinically relevant interpretations.

Suggestion: Consider organizing data in visual heatmaps (e.g., PCI vs. longevity by manufacturer and device type) to enhance accessibility.

4. The manuscript occasionally drifts into technical detail at the expense of clinical utility. While technically sound, the discussion should better highlight how clinicians could practically use PCI in device selection and programming.

Suggestion: Develop a decision-support table or tool prototype for clinicians, ideally as a supplementary file, demonstrating how PCI could inform real-world decisions.

5. The manuscript discloses relevant conflicts of interest transparently. However, due to multiple affiliations with device manufacturers, independent validation or collaboration with a neutral third party would improve perceived impartiality.

Minor Points

1. The PCI formula and its derivation should be described earlier in the text and more intuitively (e.g., through a flowchart).

2. Several instances of placeholders (“Error! Reference source not found.”) remain and must be corrected.

3. Some essential figures referenced (e.g., Figures 1–3) lack clarity or are difficult to interpret without captions. Ensure all figures are embedded with standalone readability.

4. Maintain consistency between device types (e.g., VVI vs. leadless SR), and clarify acronyms at first mention.

5. Clarify rationale behind simulation parameters, especially pacing thresholds and outputs. Are they derived from actual patient data?

Suggested Additional Experiments

To enhance the robustness and applicability of the manuscript, the following experimental extensions are recommended:

1. Apply PCI in a prospective cohort where device programming and longevity outcomes are tracked, validating its predictive capacity beyond historical data.

2. Conduct benchtop measurements for key programming features (e.g., automatic capture management, aCRT) to empirically validate modeled current assumptions.

3. Analyze how real-time remote monitoring logs correlate with PCI-predicted consumption, offering a feedback loop for future model refinement.

4. Include example cases where device selection changed based on PCI, contrasting them with conventional choice outcomes.

6. PLOS authors have the option to publish the peer review history of their article (what does this mean? ). If published, this will include your full peer review and any attached files.

**Do you want your identity to be public for this peer review?** For information about this choice, including consent withdrawal, please see our Privacy Policy .

Reviewer #1: No

Reviewer #2: **Yes: ** Sina Kazemian

Reviewer #3: **Yes: ** Tien Hoang Anh

---

## [Author Response · Author response to Decision Letter 1]

5 Aug 2025

We have uploaded a comprehensive responses document and answered the comments of the editorial board in the covering letter.

---

## [Decision Letter · Decision Letter 1]

10 Sep 2025

Cardiac Implantable Electronic Devices longevity: A novel modelling tool for estimation and comparison.

PONE-D-25-20064R1

Dear Dr. Witte,

We’re pleased to inform you that your manuscript has been judged scientifically suitable for publication and will be formally accepted for publication once it meets all outstanding technical requirements.

Kind regards,

Hamed Tavolinejad

Academic Editor

PLOS ONE

Additional Editor Comments (optional):

Reviewer #1:

Reviewer #2:

Reviewers' comments:

Reviewer's Responses to Questions

**Comments to the Author**

1. If the authors have adequately addressed your comments raised in a previous round of review and you feel that this manuscript is now acceptable for publication, you may indicate that here to bypass the “Comments to the Author” section, enter your conflict of interest statement in the “Confidential to Editor” section, and submit your "Accept" recommendation.

Reviewer #1: All comments have been addressed

Reviewer #2: All comments have been addressed

2. Is the manuscript technically sound, and do the data support the conclusions?

Reviewer #1: Yes

Reviewer #2: Yes

3. Has the statistical analysis been performed appropriately and rigorously? 

Reviewer #1: Yes

Reviewer #2: Yes

4. Have the authors made all data underlying the findings in their manuscript fully available?

Reviewer #1: Yes

Reviewer #2: Yes

5. Is the manuscript presented in an intelligible fashion and written in standard English?

Reviewer #1: Yes

Reviewer #2: Yes

6. Review Comments to the Author

Reviewer #1: I appreciate the authors’ revision in response to my questions.

They have presented very interesting work regarding CIED battery consumption.

Reviewer #2: Thank you for your detailed responses and for addressing all of my comments. I have no further comments at this stage.

7. PLOS authors have the option to publish the peer review history of their article (what does this mean? ). If published, this will include your full peer review and any attached files.

**Do you want your identity to be public for this peer review?** For information about this choice, including consent withdrawal, please see our Privacy Policy .

Reviewer #1: **Yes: ** Kentaro Goto

Reviewer #2: **Yes: ** SINA KAZEMIAN

---

## [Editor Report · Acceptance letter]

PONE-D-25-20064R1

PLOS ONE

Dear Dr. Witte,

I'm pleased to inform you that your manuscript has been deemed suitable for publication in PLOS ONE. Congratulations! Your manuscript is now being handed over to our production team.

Kind regards,

on behalf of

Dr. Hamed Tavolinejad

Academic Editor

PLOS ONE